# Study of the Genetic Expression of Antiretroviral Restriction Factors and Acute Phase Proteins in Cattle Infected with Bovine Leukemia Virus

**DOI:** 10.3390/pathogens12040529

**Published:** 2023-03-29

**Authors:** Ana S. González-Méndez, Jorge L. Tórtora Pérez, Edith Rojas-Anaya, Hugo Ramírez Álvarez

**Affiliations:** 1Virology, Genetics and Molecular Biology Laboratory, Faculty of Higher Education Cuautitlan, Veterinary Medical School, Campus 4, National Autonomous University of Mexico, Cuautitlan Izcalli, Mexico City CP 54714, Mexico; anss_silvana@comunidad.unam.mx (A.S.G.-M.); tortora@unam.mx (J.L.T.P.); 2Pacific Center Research Center, INIFAP, Guadalajara CP 44660, Mexico; edith_ra23@hotmail.com

**Keywords:** APOBEC, HEXIM, haptoglobin, serum amyloid A, BST2

## Abstract

The goal of this study was to analyze the genetic expression of antiretroviral restriction factors (ARF) and acute phase proteins (APP), as well as their correlation with proviral and viral loads in cattle with aleukemic (AL) and persistent lymphocytosis (PL). Complete blood samples were collected from a herd of dairy cows, and we extracted genetic material from peripheral blood leukocytes. Absolute quantification of the expression of ARF (APOBEC-Z1, Z2, and Z3; HEXIM-1, HEXIM-2, and BST2) and APP (haptoglobin (HP), and serum amyloid A (SAA)) was performed by qPCR. Statistical significance was observed in the expression of APOBEC-Z3 in BLV-infected animals. We only found positive correlations with a strong expression of the ARF genes in the AL group. The participation of APOBEC (Z1 and Z3), HEXIM-1, and HEXIM-2 was more frequently identified in BLV-infected animals. HEXIM-2 showed active gene expression in the AL group. Although the expression of ARF in early stages of infection (AL) maintains an important participation, in late stages (PL) it seems to have little relevance.

## 1. Introduction

Bovine Leukemia Virus (BLV) belongs to the order *Ortevirales*, family *Retroviridae*, subfamily *Orthoretrovirinae*, and the *Deltaretrovirus* genus [1]. It is an oncogenic retrovirus and the causative agent of enzootic bovine leukosis, a contagious lymphoproliferative disease of cattle. Studies indicate that BLV infection mainly generates three infection phases: persistent lymphocytosis (PL) occurs in 30% of infected cattle, asymptomatic or aleukemic infection (AL) occurs in over 60% of animals, and only 5 to 10% of infected animals develop lymphosarcomas [2,3]. Research strategies to delve into the pathogenic and immunological mechanisms of the virus have primarily focused on determining proviral load [4,5], the presence of PL and humoral or cellular immune responses [6], and the identification of genetic resistance/susceptibility factors based on the BoLA gene [7,8,9]. However, the participation of innate immune responses in BLV infection has not been studied much. Innate immunity does not require specific recognition, processing, and presentation of infectious agents to trigger a response. The response is mediated by the interaction of pathogen associated molecular patterns (PAMPs) and pattern recognition receptors (PRRs), which are primarily found on the surface of macrophages and dendritic cells [10,11]. Antiretroviral restriction factors (ARFs) are proteins involved in blocking retroviral replication, expressed by type 1 interferon-stimulated genes (ISGs—including apolipoprotein B mRNA-editing enzyme catalytic peptide 3 (APOBEC3) and bone marrow stromal cell antigen 2 (BST2), also called Teterin) and hexamethylene-bis-acetamide-inducible protein 1 (HEXIM-1), which can inhibit the retroviral replication cycle [10,11].

One of the APOBEC functions is to deaminate viral RNA, generating C to U changes and inhibiting reverse transcription and proviral integration [12]. Three APOBEC isotypes (Z1, Z2, and Z3) have been identified in cattle, with demonstrated participation in retroviral infections related to bovine immunodeficiency virus (BIV) and Jembrana disease virus (JEV) [13]. Meanwhile, BST2 is a protein capable of trapping virions inside cells, preventing the escape of viral particles and therefore preventing the infection of other cells [10]. HEXIM-1 is a little-studied protein that interacts with LTR in the transcription process in BIV infection, competing with the Tat protein for binding to cyclin T1 [10,14].

Acute phase proteins (APPs) belong to a heterogeneous group of proteins and polypeptides that constitute the first line of defense shortly after pathogen invasion. They immunomodulate both pro- and anti-inflammatory immunological mechanisms [15]. Haptoglobin (HP) and serum amyloid A (SAA) have been shown to play a role in viral infections such as caprine arthritis encephalitis, border disease, sheep and goat plague, bluetongue, bovine viral diarrhea, foot-and-mouth disease, and bovine respiratory syncytial virus (BRSV) [15,16,17]. The HP response has been found to correlate with the severity of clinical signs in animals infected with BRSV [16].

Given this context of antiretroviral and acute phase restriction protein participation in viral infections in bovines, the goal of this study was to analyze the genetic expression of ARF and APP in aleukemic infected bovines (AL) and with persistent lymphocytosis (PL), as well as its correlation with BLV viral and proviral load.

## 2. Materials and Methods

### 2.1. Study Animals

The study population included female Holstein–Friesian cattle from an intensive 1800-head production system. All animals were vaccinated against bovine respiratory disease complex (BRDC), and the herd is monitored monthly to identify *Brucella* sp. and mycobacteria infections, as well as subclinical mastitis (8% incidence). We sampled second-calving animals with an average age of 3 years. Each of the three samplings (S1, S2, and S3) were one month apart, in order to classify persistent lymphocytosis. In the present study, only two samplings (S1 and S3) were evaluated. One hundred thirteen animals were used, of which only 33 made up the final group.

### 2.2. Obtaining PBLs and Plasma

We collected blood samples through coccygeal vein puncture into tubes with anticoagulant (BD Vacutainer^®^ with Lithium Heparin, Becton Dickinson, Cuautitlán Izcalli, Edo de Mex., México). Samples were centrifuged at 350 g for 15 min for phase separation, plasma, and peripheral blood leukocytes (PBLs). Plasma was collected in microtubes, and we then processed the white layer using lysis solutions [18,19] to obtain PBLs. Samples were stored at −70 °C until use.

### 2.3. Detection of BLV Serological Infection

To determine serological condition in study animals, we tested for the presence of BLV antibodies in plasma using commercial Bovine Leukemia Virus Antibody Test Kit, which is an enzyme-linked, immunosorbent assay (ELISA) (VMRD, Pullman, WA, USA), following the manufacturer’s instructions.

### 2.4. Lymphocyte Count

Lymphocyte counts were performed on all study animals (BLV seronegative and seropositive) [20] and classified according to previously established parameters [21].

### 2.5. Nucleic Acids

Genetic material (DNA and RNA) was extracted using the protocol described by Cerkovnik et al., 2007 [22], from PBL samples using commercial reagent Trizol™ (Invitrogen, Thermo Fisher Scientific, Waltham, MA, USA) and following the manufacturer’s instructions. We quantified DNA and RNA in a nanodrop at 260−280 nm absorbance (Thermo Fisher Scientific). The RNA samples were treated with a DNase I from the Thermo Scientific RapidOut DNA Removal Kit (Thermo Fisher Scientific), following the manufacturer’s instructions. The genetic material was stored at −70 °C until use.

### 2.6. cDNA

cDNA synthesis was performed from RNA (between 50 and 100 ng/µL) with a RevertAid Synthesis Kit (Thermo Scientific, Vilnius, Lithuania), following the manufacturer's instructions, obtaining an average concentration of 2000 ng/uL, leaving a final working concentration of 200 ng/µL of cDNA.

### 2.7. Multiplex RT-PCR to Identify Bovine Respiratory Complex Viruses

We evaluated samples using the multiplex RT-PCR technique to identify bovine respiratory syncytial virus (BRSV), bovine parainfluenza virus 3 (BPIV-3), bovine herpesvirus type 1 (BoHV-1), and bovine diarrhea virus (BVDV) participants of BRDC. For the reaction mixture, we used the OneStep RT-PCR (Qiagen, Hilden, Germany) kit with the following concentrations: 1× buffer with 1.5 mm of MgCl_2_, 1 μL of enzyme mixture (reverse transcriptase and DNA polymerase), 300 nm of dNTP, DNA (300 ng/µL) and RNA (150 ng/µL), and 600 nm of each BRDC virus primer (BoHV-1, BPIV-3, BRSV, and BVDV) (see Appendix A) [22,23]; samples were amplified in 1 cycle starting with a cDNA step at 50 °C for 30 min and a denaturation step at 95 °C for 15 min, followed by 35 cycles of denaturation at 95 °C for 30 s, annealing at 59 °C for 30 s, extension at 72 °C for 40 s, and a final extension cycle at 72 °C for 15 min. For visualization, PCR products were separated by electrophoresis on 2.5% agarose gels stained with ethidium bromide (5 μg/mL), and a base pair marker was produced as a reference (Cleaver Scientific, Rugby, UK). Electrical current at 90 V was applied for one hour for subsequent visualization under ultraviolet light in a transilluminator (UVP^®^, Upland, CA, USA). All samples positive for BRDC in at least one of the samplings were discarded from the study group.

### 2.8. PCR Detection of BLV Provirus

All samples were tested for BLV infection using end-point PCR with specific primers that hybridize to the *env* gene. Used primers are shown in Appendix A (Fw2 env and Rv2 *env*) and following the protocol of Ceron et al. [19].

### 2.9. Sequencing and Phylogeny

Positive amplicons (BLV PCR) were purified using a commercial kit (Favorgen, Bioech Corp., Pingtung, Taiwan) and sent for bidirectional sequencing using Sanger’s method at the Biotechnology and Prototype Unit of FES- Iztacala, UNAM. The fragment used for the phylogenetic analysis was 477 bp of the BLV transmembrane region. The tree was built with the GENEIOUS^®^ 11.1.5, USA program using the Neighbor-Joining inference, and the statistical confidence of the topology of the phylogenetic tree was secured with bootstrap values of 1000 repetitions. FigTree^®^ v1.4.3. was used to edit the tree.

### 2.10. Sample Classification

Once animal status and cell count were established, samples were grouped into negative (N) and infected by BLV, including both aleukemic (AL) and with persistent lymphocytosis (PL).

### 2.11. Design of Primers and Positive Controls to Amplify ARF and APP

A panel of primers (Appendix A) and three synthetic genes (gBloks) were used that included the flanking regions of each primer pair designed for ARF and APP amplification, which were used as positive controls to determine copy number, and these were sent to a commercial company for synthesis. We used available sequences in GenBank for the different genes of interest: APOBEC-Z1 gene ID: 507162, APOBEC-Z2 gene ID: 504505, APOBEC-Z3 gene ID: 108771180, BST2 gene ID: 100298356, HEXIM-1 ID: 539696, HEXIM-2 ID: 614679, Serum Amyloid A ID: 104968478, and Haptoglobin gene ID: 280692.

We used Primer3, Primer-Blast, BioEdit, and Geneious bioinformatics packages and used the NEBioCalculator v1.13.1 website (https://nebiocalculator.neb.com/#!/dsdnaamt accessed on 2 March 2021) to analyze the gBlok, calculating the number of copies/µL in the reconstituted fragment. Additionally, they were evaluated with previously described formulas [24,25].

### 2.12. Evaluation of Proviral and Viral Loads

Proviral and viral loads were evaluated from DNA and cDNA, respectively, using real-time PCR (qPCR) with primers specific to the *env* region of the viral genome (Appendix A) and the Maxima SYBR Green qPCR Master Mix kit (2x) (Thermo Scientific^®^, Waltham, MA, USA). These were carried out in a 10 µL reaction volume containing 0.4 µM of primers and 1 µL of cDNA (200 ng/µL) or DNA (≥100 ng/µL), using the following amplification conditions: preincubation at 95 °C for 5 min, followed by 40 cycles at 95 °C 30 s denaturation, 62 °C 30 s annealing, and 72 °C 20 s extension. To carry out the absolute quantification of viral and proviral loads, we established the expression curve using a plasmid that included the BLV *env* region. For the BLV *env* positive control, 5 μL of plasmid DNA was used in 45 μL of molecular biology grade water to obtain 9.36 × 10^8^ copies/μL. We consider this as the first standard (Std.1). Subsequently, serial tenfold dilutions of standard 1 were made to make standards from 2 to 5 (std2, std3, std4, std5) obtaining concentrations of 9.36 × 10^7^, 9.36 × 10^6^, 9.36 × 10^5^, 9.36 × 10^4^, and 9.36 × 10^3^, respectively. It is worth mentioning that 10 serial tenfold dilutions were evaluated to establish these 5 standard points, and the copy number calculations were performed on the NEBioCalculator v1.13.1 website (https://nebiocalculator.neb.com/#!/dsdnaamt, accessed on 2 March 2021).

### 2.13. Determination of ARF and APP Expression

We determined the expression of ARF and APP with absolute qPCR in triplicate, using the Maxima SYBR Green qPCR Master Mix (2x) kit (Thermo Scientific^®^) in a 10 µL reaction volume, with 0.4 µM of primers and 1 µL of cDNA (200 ng/µL). Amplification conditions were: preincubation at 95 °C for 5 min, followed by 40 cycles at 95 °C 30 s denaturation, 54 °C 30 s annealing, and 72 °C 20 s extension. Conditions and concentrations were the same for all primers.

### 2.14. Housekeeping

The reference genes were quantified in cDNA samples using real-time PCR (qPCR) with primers specific to amplify a fragment of the TATA box-binding protein (TBP) gene and Hypoxanthine phosphoribosyltransferase 1 (HPTR-1) (Appendix A) and the Maxima SYBR Green qPCR Master Mix kit (2x) (Thermo Scientific^®^). These were carried out in a 10 µL reaction volume containing 0.4 µM of primers and 1 µL of cDNA (200 ng/µL) using the following amplification conditions: preincubation at 95 °C for 5 min, followed by 40 cycles at 95 °C 20 s denaturation, 58 °C 20 s annealing, and 72 °C 20 s extension.

### 2.15. Statistical Analysis

All data distributions were tested for normality using a Shapiro–Wilk normality test, and we also used the Bartlett test to determine data homoscedasticity (Appendix A) before performing the statistical analyses. To compare inferential statistics, we used the statplot tool in R. We used nonparametric Kruskal–Wallis tests for comparison between groups and samplings and then Dunn’s test for pairwise comparisons between each independent group to find differences between them; *p* values < 0.05 were considered significant with a 95% confidence interval. Boxplots were graphed using the ggplot R package [26]. We also used the Spearman test to evaluate the correlation between ARF gene expression (APOBEC-Z1, APOBEC-Z2, APOBEC-Z3, HEXIM-1, HEXIM-2, and BST2), the APPs (HP and SAA), number of lymphocytes, and viral and proviral loads in the different groups (AL, PL, and N). All analyses were performed using commercial R software, and *p* values < 0.05 were considered significant.

## 3. Results

### 3.1. Study Group

Animals that tested positive for BLV via serology, PCR, and qPCR were grouped according to lymphocyte number. Animals with a lymphocyte count greater than 10,000 lymphocytes/mm^3^ were placed in the PL group, those with an average count of 16,535.00 lymphocytes per mm^3^ were placed in S1 (Appendix A), and animals placed in S3 (Table 1 and Appendix A) had an average count of 18,931.60 lymphocytes/mm^3^. The AL group contained animals that tested positive for BLV and had an average S1 count (Appendix A) of 4905.39 lymphocytes/mm^3^ and an average of 7566.19 lymphocytes/mm^3^ for S3 (Table 1 and Appendix A). The BLV-negative group had an average lymphocyte count of 3164.78 lymphocytes/mm^3^ for S1 (Appendix A) and an average of 5359.67 lymphocytes/mm^3^ for S3 (Table 1 and Appendix A). One hundred thirteen animals were used that were sampled three times with 1-month intervals between each sampling; each sampling was evaluated applying the exclusion criteria. The final study nucleus was made up of 33 bovines distributed across 16 AL bovines, 9 with BLV-positive PL and 8 with BLV-negative (N) bovines.

### 3.2. BLV Viral and Proviral Load Determination

Proviral and viral load quantification (number of copies/µL) was performed for samples 1 (S1) and 3 (S3). The proviral load in the AL group had a mean value of 5.09 × 10^5^ (S1: 6.16 × 10^5^, S3: 4.02 × 10^5^ (Table 1 and Appendix A)), while the PL group had an average of 2.63 × 10^6^ (S1: 2.37 × 10^6^, S3: 2.90 × 10^6^ (Table 1 and Appendix A)). The viral load for the AL group showed an average value of 4.48 × 10^3^ (S1: 8.37 × 10^2^, S3: 8.12 × 10^3^ (Table 1 and Appendix A)), while the PL group had an average of 2.70 × 10^4^ (S1: 5.04 × 10^3^, S3: 4.89 × 10^4^ (Table 1 and Appendix A)). It is worth noting that no viral load was detected in four of the samples from the PL group.

### 3.3. Genotyping

Amplicons obtained in the BLV-*env* PCR were purified and sent for sequencing by the Sanger method for both chains, resulting in ten nucleotide sequences from the study animals, nucleotide sequences were deposited in GenBank, and are available with access numbers OQ190824 to OQ190833. Fragments of a length of 477 bp were used to build a phylogenetic tree which included reference sequences to determine the infecting BLV genotype. All study sequences were associated with genotype 1 (Appendix A).

### 3.4. Housekeeping

When performing the housekeeping analysis, DNA contamination was identified in the samples, and therefore they were treated with DNase I. The values of the threshold cycle (CT) obtained in the expression of housekeeping (TBP) oscillate between 33.47 to 38.10, with an average of 35.60. The CTs obtained in the expression of Hypoxanthine phosphoribosyltransferase 1 (HPRT-1) oscillate between 33.53 and 43.82 with an average of 36.30. In general, in 88% of the samples, values were identified that did not exceed five CTs of difference between them, and only four samples had CT values that were too large.

### 3.5. ARF and APP Gene Expression

We used LightCicler Roche 96 software (Version 1.1.0.1320) to quantify samples and qPCR efficiency. The R2 values for qPCR quantification curves ranged between 0.97 and 1.00 (Appendix A), and quantification values (number of copies/µL) for each gene are shown in Appendix A and in Figure 1. Detection of ARF and APP gene expression was carried out for S1 and S3, and we found the highest APOBEC-Z1 expression (6.50 × 10^6^) in the S3 sampling of the BLV-negative population, followed by the AL group (6.64 × 10^5^) of the S3 sampling. The APOBEC-Z2 expression was identified in both samplings from cattle infected with BLV (AL 3.63 × 10^2^ and PL 1.04 × 10^1^); however, it was only expressed in four samples. The highest S3 APOBEC-Z3 expression was identified in infected animals (AL 6.20 × 10^3^ and PL 7.14 × 10^2^) and showed a significant value in the AL and PL groups (Appendix A).

Generally, we found higher expression of BST2 and HEXIM-1 in cattle infected with BLV, and the highest expression of BST2 (8.74 × 10^2^) and HEXIM-1 (2.05 × 10^1^) was in S3 of the AL. Similarly, HEXIM-2 expression was higher in cows infected with BLV (AL 6.37 × 10^1^) in S3. We also found the highest APP expression values in S3. Particularly in the BLV-negative population, SAA was quantified at 1.55 × 10^7^ and in cattle infected with BLV (AL 4.23 × 10^4^ and PL 4.23 × 10^3^); data from the statistical analysis are shown in Appendix A and Appendix A. In the study population, the expression of HP was not determined, with the exception of one BLV-positive bovine (AL). The expression of APOBEC-Z1 was identified in all the study population, while animals with the highest expression of APOBEC-Z3 were positive for BLV. In AL cows, the highest expression was identified in the BST2, HEXIM-1, and HEXIM-2 genes. We found that SAA was expressed more in animals that were not infected with BLV.

We analyzed the correlation of lymphocyte count, viral load, proviral load, and the ARF and APP genes (Table 1 and Appendix A), and Figure 2 shows the main findings. When determining ARF and APP participation in BLV-infected cattle (AL and PL), most of the identified gene expression correlations were positive, indicating that the expression is directly proportional among the correlated genes. We only found a negative correlation in the PL group (APOBEC Z1 with Z3), indicating an inversely proportional expression. Our analysis showed a series of expression correlations between S3, ARF, and SAA in cows infected with BLV (AL and PL) (data shown in Table 1 and Appendix A).

Overall, we found that the most important gene expression correlations in the AL group were with proviral and viral load, APOBEC Z1 and Z3, BST2, HEXIM-1, HEXIM-2, and SAA. In contrast, the most frequent gene expression correlations in the PL group were with proviral load, viral load, and lymphocyte counts; in addition, correlations were identified among APOBEC Z1 with Z3, APOBEC Z3 with BST2, and SAA with HEXIM-1 (Figure 2); however, the values were low compared to those identified in AL cows.

## 4. Discussion

The antiretroviral factors (ARF) gene expression results showed that APOBEC Z1 and APOBEC Z3 were the genes with the highest statistical relevance as well as the greatest participation in the analyzed correlations. Although some ARF expression was identified in animals not infected with BLV and with a low level of expression, it is possible that BLV-negative cattle had a retroviral infection not detected by the tests used in the present study. Infections associated with the bovine immunodeficiency virus (BIV) in Mexican cattle have been described [28]. The antiviral effect of the APOBEC protein is to deaminate viral RNA, generating C to U changes, which inhibit reverse transcription and proviral integration [12]. APOBEC decreases viral infectivity of the retrovirus responsible for Jembrana disease virus (JDV) and bovine immunodeficiency virus (BIV) [29]. The function of APOBEC on RNA has been widely described in retroviruses and has also been described for families of DNA viruses [30]. Specifically, in the case of the *Deltaretrovirus* genus, the mechanisms of action associated with APOBEC are the hypermutation and inactivation of Tax, and its antagonist is the viral nucleocapsid [30].

Three APOBEC isotypes (Z1, Z2, and Z3) have been identified in bovines, and it has been shown that the structure of bovine APOBEC-Z3 is similar to human APOBEC3H [13,31] and that APOBEC3H is the most powerful replication inhibitor for the human immunodeficiency virus (HIV) [13]. It is important to mention that BLV belongs to the *Deltaretrovirus* genus and does not express the viral infectivity factor (*Vif*), which is encoded by the *Lentivirus* genus; this factor has been shown to be an APOBEC antagonist [29]. However, APOBEC participation has been shown in other *Deltaretrovirus*, such as Human T-cell Leukemia Virus type 1 (HTLV-1). The APOBEC antagonist mechanism of HTLV-1 is a peptide motif in the C-terminal domain nucleocapsid (NC) protein that acts in *cis* to inhibit APOBEC3G packaging [30]. This could explain why the expression of the different APOBEC isotypes (Z1 and Z3) correlated with one negative expression in the persistent lymphocytosis (PL) group (Figure 2 and Appendix A). This lack of correlation could be associated with the inhibitory effect of APOBEC-Z3 related to a higher viral load (nucleocapsid) that was identified in the PL group, although the antagonistic function of BLV against APOBEC has not been described.

It is important to mention that the number of lymphocytes counted in AL animals (average 6235.79) was close to those counted (average 4262.22) in animals negative to BLV. However, we only identified strong positive correlations in the different APOBEC (Z1 and Z3) with other ARFs in the AL group, indicating that it is related to the presence of BLV (Figure 2). Other studies have associated APOBEC3G as a new biomarker in ovarian cancer prognosis in tumor-infiltrating T-lymphocytes [32]. Although it is possible to identify tumor phases in BLV infections, this group of animals was not included in this study. However, it has been demonstrated that animals that develop the PL phase could progress to tumor phases of the disease [33,34]. The lack of correlations in APOBEC gene expression, viral and proviral loads, may be indicative of the transition between infection phases (AL to PL).

HEXIM-1 acts as a tumor suppressor regulating the p53 pathway [35]. It participates in activation of the innate immune response, differentiation, development, and inflammation [36], and it also inhibits the elongation factor b positive transcription (P-TEFb), which controls RNA polymerase II transcription [37]. Similarly, HEXIM-II is known as a paralog protein of HEXIM-1 [38]; HEXIM-1 has been shown to be inhibited by bovine-immunodeficiency-virus-mediated *tat* transactivation. Unlike BIV, BLV lacks *tat* and the absence of this gene, given that no other antagonists for this protein have been described, likely favoring the expression of HEXIM in BLV-infected animals. In the infected cattle of the AL group, we found a higher frequency of gene expression correlations of HEXIM-1 and HEXIM-2 with APOBEC (Z1 and Z3) (Figure 2). The analyses we carried out did not allow us to observe a direct participation with respect to a greater gene expression of HEXIM with a decrease in the quantification of the viral and proviral load. Additionally, a positive correlation was identified in the expression of HEXIM-1 and BST2 in BLV-negative bovine. On the other hand, correlations were identified in the expression of HEXIM with APP in animals infected with BLV (AL and PL groups), and this may be associated with other viral infections not related to bovine respiratory complex or bacterial infections, which could not be identified in this study. Because of this, it is important to carry out in vitro studies to determine if HEXIM-1 and/or HEXIM-2 achieve efficient blocking of BLV virion production.

Studies of BST2 are based on functionality and blocking the release of viruses that can infect humans and other species; however, little information exists regarding viral infections in bovines. Taqueda et al. (2012) showed a link between in vitro expression of bBST-2A1 or bBST-2A2 and reduced production of BLV viral particles and vesicular stomatitis virus [39]. This association was not observed in the present study, the expression of BST2 being correlated with the expression of APOBEC Z1 and Z3 in bovines infected with BLV, which suggests that the same stimulus could be favoring the expression of these ARFs; however, gene expression of these proteins is not efficient in reducing virus production in PL bovines, which could be explained by the mechanism used by BLV in its propagation. In animals with PL, virus propagation has been documented through clonal expansion of infected cells [40], similar to mechanisms described for other *Deltaretroviruses* such as HTLV-1 [41,42]. In AL animals, a higher frequency was observed in the correlation of APOBEC (Z1 and Z3) with BST2, and although no negative correlations were identified between these ARF and viral load, a lower viral load (4.48 × 10^3^) was determined, which could be related to the expression of APOBEC and BST2 in this group of animals or to the propagation of de novo BLV in susceptible cells.

Serum amyloid A (SAA) protein is a normal constituent of blood serum, it is mainly- but not exclusively synthesized in the liver, and it is the protein with the greatest participation in the acute phase response. The biological functions of SAA are unresolved, but its traits are consistent with a prominent role in primary host defense [43], and SAA may have an important role in BLV-related infections. However, in this study, we identified an SAA gene expression correlation in all the study groups, including animals not infected with BLV. This may be associated with other viral infections not related to bovine respiratory complex or bacterial infections, which could not be identified in this study. Heegard et al. (2000) found higher SAA expression during acute phases in experimental respiratory syncytial virus (RSV) infections. It is noteworthy that many studies of APPs in viral infections [15,16,44,45,46] have identified the participation of SAA and HP. In these studies, SAA is frequently identified during acute phases of viral infections, while HP is more frequently correlated with chronic infections, which, in turn, are also linked to greater severity of clinical signs [16,47,48].

HP is an acute phase polymorphic protein that is produced in the liver [49]. In healthy bovines, it may be absent or found at low concentrations that increase rapidly in response to infections [50]. Höfner et al. (1994) detected HP in bovines naturally infected with foot-and-mouth disease virus, and they found a significant increase in HP in serum after the onset of viraemia and the appearance of clinical signs [50]. We did not identify the expression of HP in any of the three study groups.

The reference genes used in this study were previously described by Brym et al. (2013) [51], considering that they showed a stable expression in bovines. However, in this study, some samples were identified with very wide CT values; nevertheless, this did not affect the expression values of ARF and APP, considering that the quantification was absolute using synthetic genes. On the other hand, it is necessary to conduct more analysis to correctly choose reference genes [52] for gene expression studies in bovines.

BLV proviral load determination methodologies are variable. Some use in vitro methods to detect the *pol* gene [53], other designs are based on LTR detection using qPCR and the Coordination of Common Motifs (CoCoMo) algorithm [54], in addition to using commercial kits [55] or simply using predictive models [56]. We performed an absolute qPCR quantification based on detection of the BLV *env* gene, without determining the number of infected cells as other studies have accomplished. Because of this, the proviral load was not classified as high or low. Although our data cannot be compared with data from other studies, we were able to identify a 5.18-fold increase in proviral load in the PL group (26,742,286.66) compared to the AL group (509,333.0). This higher PL proviral load did not correlate with the number of lymphocytes. Lastly, few studies have determined the BLV viral load. While in our study the BLV viral load was lower than the proviral load, both values maintained a similar proportion (6.02 times) between the two groups of infected bovines, being higher for the group PL (27,012.80) (Table 1).

## 5. Conclusions

While ARF and APP gene expression does not seem to influence either the proviral or viral load in cattle infected with BLV genotype 1, we found an important genetic expression of different ARF in animals infected with BLV compared to the negative group. We found that the correlation of the gene expression of APOBEC (Z1 and Z3) with different ARFs was more relevant for the AL group. The expression of APOBEC (Z1 and Z3) and HEXIM (1 and 2) was identified for the first time in BLV-infected cattle at different stages of infection. On the other hand, the correlation of the gene expression of BST2 and APOBEC (Z1) was identified in the early stages of BLV infection. SAA gene expression in peripheral blood leukocyte cells was quantified with qPCR and may be another option for identification. A methodology was established to determine the viral and proviral load based on the detection of the BLV env gene. Although the expression of ARF in early stages of infection (AL) maintains an important participation, in late stages (PL) it seems to have little relevance.

## Figures and Tables

**Figure 1 pathogens-12-00529-f001:**
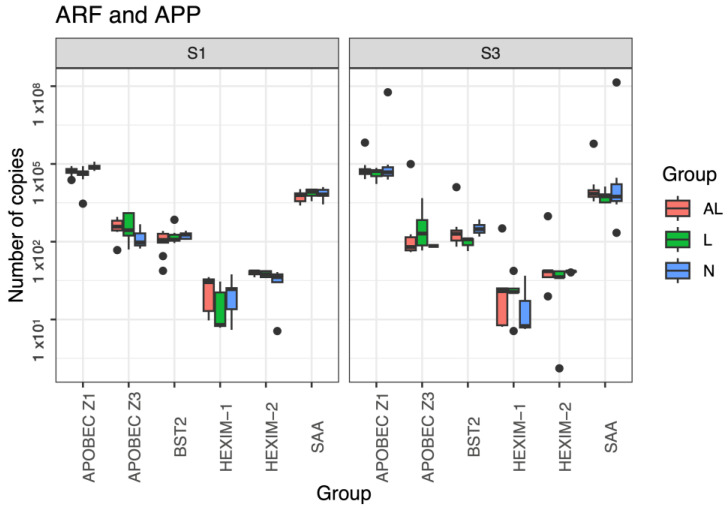
Boxplot of gene expression of antiretroviral restriction factors (ARF) and acute phase proteins (APP). Boxplot indicating the expression of the different genes evaluated, separated by sampling graph. AL (coral): group of aleukemic animals infected with BLV; PL (green): group of animals with persistent lymphocytosis infected with BLV; N (blue): group of animals not infect-ed with BLV (negatives); S1: sampling 1; S3: sampling 3; points outside boxplots indicate outline values; APOBEC: catalytic apolipoprotein B mRNA-editing enzyme (Z1 and Z3); HEXIM: inducible protein hexamethylene-bis-acetamide (1 and 2); BST2: bone marrow stromal cell antigen 2; SAA: serum amyloid A.

**Figure 2 pathogens-12-00529-f002:**
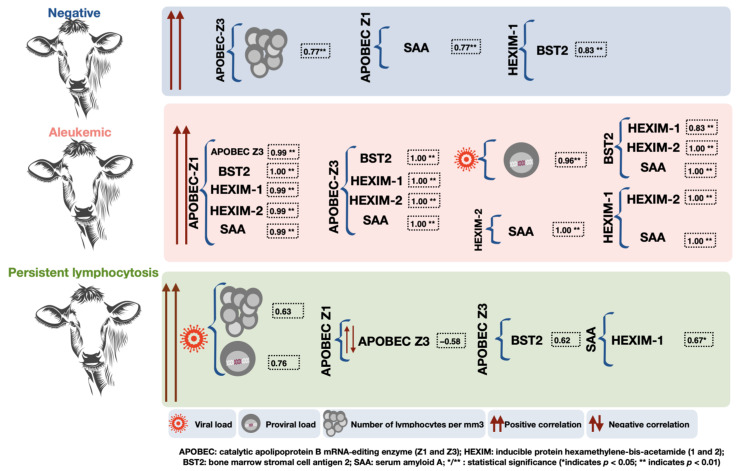
Summary figure of the expression correlations of antiretroviral restriction factors (ARF) and acute phase proteins (APP) in the study groups (aleukemic (AL), persistent lymphocytosis (PL), and negative to BLV). APOBEC Z1 and Z3 gene expression correlated with different ARFs and APPs in all study groups, including the BLV-negative group; however, the AL group had the highest number of correlations between ARF and APP, additionally showing a positive correlation between viral and proviral load. The correlations identified between ARF and APP in the PL group were moderately positive, showing that viral load correlated positively with both lymphocytosis and proviral load.

**Table 1 pathogens-12-00529-t001:** Summary of ARF and APP gene expression from bovines negative to BLV (N) and infected with BLV, aleukemic (AL), and with persistent lymphocytosis (PL). Sampling analysis 3 (S3).

Variable	Group	M	SD	L⌀’s	Proviral	APOBECZ1	APOBECZ3	BST2	HEXIM1	HEXIM2
Lymphocytes (L⌀’s)	N	5359.67	1223.49							
	AL	7566.19	2596.64							
	PL	18,931.60	6338.98							
Proviral	N	0.00	0.00							
	AL	402,277.00	696,725.51							
	PL	2,905,680.00	5,060,288.29							
Viral	N	0.00	0.00							
	AL	8123.06	14,127.92		0.96 ** [0.90, 0.99]					
	PL	48,980.50	47,596.10	0.63 [−0.00, 0.90]	0.76 * [0.25, 0.94]					
APOBECZ1	N	6,504,834.44	19,381,955.65							
	AL	91,033.12	153,826.40							
	PL	43,547.00	22,684.39							
APOBECZ3	N	7.64	22.92	0.77 * [0.23, 0.95]						
	AL	6197.56	24,683.37			0.99 ** [0.98, 1.00]				
	PL	713.56	1478.42			−0.58 [−0.89, 0.07]				
BST2	N	167.41	250.72							
	AL	874.32	3155.61			1.00 ** [0.99, 1.00]	1.00 ** [1.0, 1.0]			
	PL	29.84	53.21				0.62 [−0.02, 0.90]			
HEXIM1	N	0.55	1.62					0.83 ** [0.38, 0.96]		
	AL	20.53	81.35			0.99 ** [0.98, 1.00]	1.00 ** [1.0, 1.0]	1.00 ** [1.0, 1.0]		
	PL	1.14	2.32							
HEXIM2	N	3.14	3.73							
	AL	63.66	239.35			0.99 ** [0.98, 1.00]	1.00 ** [1.0, 1.0]	1.00 ** [1.0, 1.0]	1.00 ** [1.0, 1.0]	
	PL	3.48	3.24							
SAA	N	15,472,451.06	46,397,831.76			1.00 ** [1.0, 1.0]				
	AL	42,266.75	149,198.29			0.99 ** [0.98, 1.00]	1.00 ** [1.0, 1.0]	1.00 ** [1.0, 1.0]	1.00 ** [1.0, 1.0]	1.00 ** [1.0, 1.0]
	PL	4232.30	4169.27						0.67 * [0.06, 0.91]	

M and SD are used to represent mean and standard deviation, respectively. Values in square brackets indicate the 95% confidence interval for each correlation. The confidence interval is a plausible range of population correlations that could have caused the sample correlation [27]. * indicates *p* < 0.05; ** indicates *p* < 0.01.

## Data Availability

Nucleotide sequences were deposited in GenBank, and are available with access numbers OQ190824 to OQ190833.

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
