# Peer review of "Study of the Genetic Expression of Antiretroviral Restriction Factors and Acute Phase Proteins in Cattle Infected with Bovine Leukemia Virus"

_pathogens, 2023, doi:10.3390/pathogens12040529_

Round 1

Reviewer 1 Report

I found the paper of Ana Silvia González-Méndez et al is very interesting, but difficult to use. I believe that the table 1, 2 and 3 should be moved to additional materials and replaced with a summary table of the results obtained. Moreover, listing the statistical data in the results does not facilitate reading! Although the work done by the authors is undeniably a lot, but the only clear scheme for me is the final summary (Figure2)!

Reviewer 2 Report

In the manuscript authors described expression of antiretroviral restriction factors and acute phase protein and their correlation with bovine leukemia status. The research is interesting and topic has potential and novelty. Unfortunately, the study is questionable; especially the lack of some information in the materials and methods section creates doubts about the correctness of the tests performed.

Comments:

Line 77 – Information about S2 timing cannot be found over the study?

Line 91 - It is not clear from what (plasma or PBL or both) the DNA and RNA were isolated

Line 96 - any information about DNAse treatment of RNA

-any information about amount of RNA taken to cDNA synthesis

Line 98-106 – for these reactions directly RNA were used??? What amount?

102 - spelling – Qiagen

104 – spelling – dNTP’s

105 – Table S1 – position of primers will be useful information, particularly BLV.

Line 107-109 – PCR detection of BLV provirus - whether it was a nested PCR? qPCR? What were PCR conditions and reagents? What amount of DNA per reaction?

Line 110 – what size of PCR products we used for sequencing and analysis???

What was the reason for sequencing and genotyping BLV in this study relating to expression of antiretroviral restriction factors???

Line 121 and 129 – please explain gBloks, gBlok???

Line 131-139 – what amount of DNA was used? How many nanograms? Are the authors aware that for this type of reaction the same amount of material should be taken for all samples tested?

How many repeats of done for each qPCR reactions?

What kind of dilution of plasmid was made?

In what units is the number of copies expressed - per reaction, per volume?

Line 168-170 - Please explain – the final study nucleus was made up?

Line 201 and Figure S1 – what size of fragment was used for genotyping? Why G8,G11 and G12 were not used in the analysis? Any Accession number?

More work on Deltaretoviruses and the agents studied would be useful in the discussion. When reading the paper, there is some concern as to whether the method of expression analysis used is appropriate ( what about using a reference gene or newer methods of expression testing) -  citations in support of the methodology used...as well as the approach taken will be appreciated.

Tables are not placed in the Supplementary descriptions in manuscript.

Reviewer 3 Report

The manuscript no. pathogens-2102486 by Gonzalez-Mendez A.S. et al. aims to study the genetic expression of antiretroviral restriction factors and acute phase proteins in cows infected with BLV. The topic is very interesting and worth to be studied however in my opinion presented study has not been carried out according to the standards required for gene expression studies. Therefore the paper should not be published in present form.

The main problem is the methodology which is described very chaotically and lacks informations which are required to rely on the data:

1.    Lines 72-77 how many animals were sampled and analysed in each group. In lines 169-170 authors write that there were 40 tested cows, but were the same animals tested/sampled twice or three times. There are different statements about that.

2.    Lines 96-97 – lack of information how much RNA was used for reverese transcription and whether it was treated with DNase to get rid of DNA traces which is extremely important in gene expression studies, unless the primers are located in different exons of the specific gene. But authors do not mention that.

3.    Lines 107-109 – lack of information on the PCR conditions, authors also do not refer to any other paper including such details.

4.    Lines 113-114 – why this kind of phylogentetic tree was selected?

5.    Line 121 authors describe gBlocks – please explain the idea, it would be easier for readers to follow the methodology

6.    Line 135 and 142– since authors use viral and proviral load for comparison between animals and groups and later on for statistical analysis, data need to be comparable, but authors do not mention about the concentration of DNA/cDNA used for qPCR reactions. 1 ul of DNA from diffrerent animals may contain completely diffeent amount of DNA/cDNA.

7.    Lines – 140-145 –In the gene expression study auhors do not use any reference/house-keeping gene with stable expression which is routinely used in such studies. Without such control one can not be sure whether the various gene expression observed in the samples results from the BLV infection, individual characteristic of the animal or is rather the consequence of the various amount of template used for qPCR.

Minor points:

- Results: Authors mentioned in M&M about three samplings(S1, S2, S3), while described the data only from S1 and S3.

- Lines 172-180 – This paragraph seems to be more informative when presented as graph or table, since in present form, with plenty of numbers, it is not easy to follow.

- Tables 1 to 3. The description of the tables is very poor. There is lack of information what the numbers 1-10 following M and SD stand for.

- The description in footnotes in Table 2 (lines 193-194) is confusing – black numbers should indicate the unique values of AL-group not negtive?

Round 2

Reviewer 2 Report

The authors have clarified many of the reviewer's comments; however, there are still some inaccuracies in the methodology that may affect the results of the paper.

Still, the work needs to be corrected and add possible experiments, therefore, the reviewer's position - remains at - major revision. Please see comments:

Line 96 - any information about DNAse treatment of RNA

Answer:  On line 100, the following was added: "RNA treatment with DNAses was not performed."

Due to carelessness, the treatment with DNAse I was not carried out; however, and even though we are not certain of the possible traces of DNA in the samples, there could be a possible modification in the values of genetic expression. It is important to mention that the extracted RNA samples were analyzed in agarose gels and the presence of contaminating DNA was not visualized. Additionally, there was a 260/280 reading values averaged absorbances >1.78 indicating acceptable purity. Unfortunately, most of the samples are no longer available, so it is not possible to repeat the tests with the DNAses treatment.

Also attached to line 323 is the following paragraph "The treatment of RNA with DNAse is important for this type of study; however, it was not carried out, so it is  possible  that DNA traces could modify the expression values in some of the samples. However, a differential expression was identified between the groups studied and in the analysis of the correlations.”

260/280 reading values would not show DNA traces but contamination by proteins and/or chemicals as phenol. In agarose gels is very difficult to see minor DNA contamination.

The recommended solution will be to include a no-RT qPCR control reaction to detect gDNA contamination – therefore if some samples still left this experiment should be done.

If the values for individual samples in the groups are wrong then the estimates for the group won't be quite right either.

105 – Table S1 – position of primers will be useful information, particularly BLV.

Answer: in table S1, the following text was added “ * primer FW position 6112-6130, Rv position 6192-6211; ** Fw position 5852–5871, Rv position 6506–6488, primer position was based on the OIE reference sequence with accession number K02120.”

Line 107-109 – PCR detection of BLV provirus - whether it was a nested PCR? qPCR? What were PCR conIf the values for individual samples in the groups are wrong then the estimates for the group won't be quite right either.ditions and reagents? What amount of DNA per reaction?

Answer: On lines 118 to 120 the paragraph was modified. “PCR detection of BLV provirus: All samples were tested for BLV infection using end point PCR with specific primers that hybridize to the env gene. Used primers are shown in Table S1 (Fw2 env and Rv2 env) and following the protocol of Ceron et al. [19]”

But at cited Ceron paper Fw2 and Rv2 primers have different position (5256-5275 and 5995-6012 ) then primers described at S1???

Why G8,G11 and G12 were not used in the analysis?

Answer:  the phylogenetic tree was redone including BLV genotypes 8 and 11 (Figure S1).

Genotype 12 was not considered in this study, because the sequence corresponds to another region located at position 4,831-5,738 of the BLV, while our sequences are located at position 5,907 to 6,388, according to the reference sequence. OIE K02120.

Sequence MG800834 is not G11 representative.

What about using a reference gene?

Answer: Information regarding the reference gene used in the study was included in supplementary table 5 (Table S5). In addition, information was added to lines 166- 172. “Housekeeping: The reference gene was quantified in cDNA samples, using real-time PCR (qPCR) with primers specific to amplify a fragment of the TATA box binding protein (TBP) gene (Table S1), and the Maxima SYBR Green qPCR Master Mix kit (2x) (Thermo Scientific, USA). These were carried out in a 10 µl reaction volume containing 0.4µM of primers and 1µl of cDNA (200 ng/µl), using the following amplification conditions: preincubation at 95ºC for 5 min, followed by 40 cycles at 95ºC 20 sec denaturation, 58ºC 20 sec annealing and 72ºC 20 sec extension.”

This information was not initially considered because an absolute quantification was performed using the synthetic genes; however, the determination was made and is attached to the article.

On lines 222 - 225 the following paragraph "Housekeeping" was added: “The cycle threshold (CT) values obtained in the expression of housekeeping (TBP) are shown in Table S5, although these values differ between the groups (AL 30.95, PL 29.98 and N 32.90), no statistical significance was identified between them (data not shown).”

Expression value obtained for housekeeping gene when you looked in S5 table differ even 6 even 7 Ct value between samples in one group – this difference indicated or uneven amount of RNA taken for reaction or going back to the issue in question – contamination of some samples by genomic DNA. Mean value are OK, up to value 2 of Ct but not data for individual samples.

Please, on the remaining available samples from this experiment, check another reference gene, for which the primers would be anchored in the sequence of exons separated by an intron - then the presence of a specific product of the right size and the corresponding Ct ranges will allow to judge the absence of contamination and the correctness of the reaction.

Author Response

All observations suggested by the reviewers were considered. When performing the housekeeping analysis, we identified DNA contamination in the samples, so we decided to search for the samples still available and treat them with DNase I. Once treated, new quantifications were performed and the new obtained values were statistically analyzed. With this new information, the results, discussion, conclusions, and abstract sections were restructured.

Author Response

(The authors gave the same response as above.)

Round 3

Reviewer 2 Report

The reviewer recognizes the work that the authors have put into repeating the experiments and performing a new analysis. Now there are only minor comments to make about this article, in order to improve some of the content:

1. From a recent review missed the comment - Sequence MG800834 is not G11 representative Fig.1.

2. Line 189 – this time should be 33 made up the final group - after DNAse treatment of RNA samples

3. Line 494-497 – results obtained for housekeeping genes: Should these results be seen in the context of the detected contamination of RNA samples with genomic DNA? As the Authors in their responses wrote: When performing the housekeeping analysis, we identified DNA contamination in the samples, so we decided to search for the samples still available and treat them with DNase I.

OR

Results obtained for housekeeping genes….

The fluctuations in Ct values are again too large - they are, for TBP, almost 5 Ct and for HPRT-1, as much as 10 Ct, respectively. This means that is not good choice for reference genes.

Please see just an example paper - https://journals.plos.org/plosone/article?id=10.1371/journal.pone.0141853

Please comment on this paragraph properly because a reader with little experience in this kind of analysis may misunderstand it and apply it potentially in some of his work.

Line 757-759 - it is not clearly written what is correlated with what

Line 1000 – en bovinos PL…spanish

Author Response

The reviewer recognizes the work that the authors have put into repeating the experiments and performing a new analysis. Now there are only minor comments to make about this article, in order to improve some of the content:

  1. From a recent review missed the comment - Sequence MG800834 is not G11 representative Fig.1.

Answer: The sequence MG800834 was assigned to the correct genotype (G6E). The available sequences of genotype 11 (accession numbers KU764746 and KU764747) correspond to another region located at position 5130 to 5554 of BLV, while the sequences obtained in this study are located at position 5907 to 6388, according to the reference sequence OIE K02120. For this reason, G11 sequences were not included in Figure S1.

  1. Line 189 – this time should be 33made up the final group - after DNAse treatment of RNA samples.

Answer: On line 78, Done

One hundred thirteen animals were used, of which only 33 made up the final group.

  1. Line 494-497 – results obtained for housekeeping genes: Should these results be seen in the context of the detected contamination of RNA samples with genomic DNA? As the Authors in their responses wrote: When performing the housekeeping analysis, we identified DNA contamination in the samples, so we decided to search for the samples still available and treat them with DNase I.

OR

Results obtained for housekeeping genes….

The fluctuations in Ct values are again too large - they are, for TBP, almost 5 Ct and for HPRT-1, as much as 10 Ct, respectively. This means that is not good choice for reference genes. 

Please see just an example paper - https://journals.plos.org/plosone/article?id=10.1371/journal.pone.0141853

Please comment on this paragraph properly because a reader with little experience in this kind of analysis may misunderstand it and apply it potentially in some of his work.

Answer: On Lines 218-219, the following text was included “When performing the housekeeping genes analysis, DNA contamination was identified in the samples, and therefore they were treated with DNase I.”

Answer: On Lines 223-224, the following text was included “In general, in 88% of the samples, values were identified that did not exceed 5 CTs of difference between them and only 4 samples had too large CT values”.

The wider fluctuations in CT values only occurred in 4 samples, in the evaluation of the 2 housekeeping genes, which were the lower and upper ranges that were included in the paragraph. However, the expression values in this study were not generated based on housekeeping genes, since the quantification was performed in an absolute way using synthetic genes. The chosen housekeeping genes are based on the publication Brym et al (2013) [1].

Answer: On Lines 391-396, the following paragraph was included “The reference genes used in this study were previously described by Brym et al., (2013) [1] considering that they showed a stable expression in bovines. However, in this study some samples were identified with very wide CT values; nevertheless, this did not affect the expression values of ARF and APP, considering that the quantification was absolute using synthetic genes. On the other hand, it is necessary to conduct more analysis to correctly choose reference genes [2] for gene expression studies in bovines.”

Line 757-759 - it is not clearly written what is correlated with what

Answer: On Line 271, the paragraph was restructured “APOBEC Z1 and Z3, APOBEC Z3 and BST2, SAA and HEXIM-1 (Figure 2),”

Line 1000 – en bovinos PL…spanish

Answer: On Line 360 “en bovinos PL” was changed to “in PL bovines”

  1. Brym, P.; Ruść, A.; Kamiński, S. Evaluation of Reference Genes for QRT-PCR Gene Expression Studies in Whole Blood Samples from Healthy and Leukemia-Virus Infected Cattle. Vet. Immunol. Immunopathol. 2013, 153, 302–307, doi:10.1016/j.vetimm.2013.03.004.
  2. Chapman, J.R.; Waldenström, J. With Reference to Reference Genes: A Systematic Review of Endogenous Controls in Gene Expression Studies. PLoS One 2015, 10, e0141853, doi:10.1371/journal.pone.0141853.

Reviewer 3 Report

The manuscript no. pathogens-2102486 has been reviewed according to the suggestions. The methodological errors were corrected. The presentation of the results is better now, but I still have some remarks:

1. First of all the text is difficult to follow in some fragments therefore I would suggest the English correction. There are some grammatical and stylistic errors, i.e. line 69, 70 correlate in instead of correlate with/to.

2. Please correct typos and punctuation:

lines: 264 (bovine parainfluenza);

488 Sanger metod;

873/874 Jembrana disease;

1000 in bovines;

988 and 994 in vitro should be in italics;

952 space after HTLV-1; 1114  delete the point after ref 43;

1120 please correct the sentence, delete one "participation";

1128 should be rather "...and they found increase of HP in serum after viraemia...";

1159-1161 please correct the last sentence of the conclusion, in this version it is not clear.

Author Response

The manuscript no. pathogens-2102486 has been reviewed according to the suggestions. The methodological errors were corrected. The presentation of the results is better now, but I still have some remarks:

  1. First of all the text is difficult to follow in some fragments therefore I would suggest the English correction. There are some grammatical and stylistic errors, i.e. line 69, 70 correlate in instead of correlate with/to.

Answer: On lines 24-26, the text was changed from “The data suggest that ARF gene expression is highly correlated in the early stages of BLV (AL) infection, whereas in PL animals they are less correlated.”  to “Although the expression of ARF in early stages of infection (AL) maintains an important participation, in late stages (PL) it seems to have little relevance.”

  1. Please correct typos and punctuation:

lines: 264 (bovine parainfluenza);

Answer: On Line 105 “bovine parainfluence” was changed to “bovine parainfluenza virus 3”

488 Sanger metod;

Answer: On Line 212, done

873/874 Jembrana disease;

Answer: On Lines 297 - 298 “Jambra disease” was changed to “Jembrana disease virus”

1000 in bovines;

Answer: on Line 360 "en bovinos PL" was changed to "in PL bovines"

988 and 994 in vitro should be in italics;

Answer: On Lines 348 and 354, done

952 space after HTLV-1;

Answer: On Line 311, done.

1114 delete the point after ref 43;

Answer: On Line 372, done.

1120 please correct the sentence, delete one "participation";

Answer: On Line 379, done.

1128 should be rather "...and they found increase of HP in serum after viraemia...";

Answer: On Lines 387-388, the text was changed from “and they found a significant increase in serum HP after viraemia onset and the appearance of clinical signs” to “and they found a significant increase of HP in serum after the onset of viraemia and the appearance of clinical signs”

1159-1161 please correct the last sentence of the conclusion, in this version it is not clear.

Answer: On Lines 424-426, the paragraph was restructured “Although the expression of ARF in the early stages of the infection (AL) maintains an important participation, nevertheless in late stages (PL) the expression of these ARF factors seems to be of little relevance in the late stages (PL).” was changed to “Although the expression of ARF in early stages of infection (AL) maintains an important participation, in late stages (PL) it seems to have little relevance.”